# Health Literacy, Health Behavior and States of Health among Trainee Personnel in Northern Germany

**DOI:** 10.3390/healthcare9060757

**Published:** 2021-06-18

**Authors:** Susanne Steinke, Peter Koch, Janna Lietz, Zita Schillmöller, Albert Nienhaus

**Affiliations:** 1Competence Center for Epidemiology and Health Services Research for Healthcare Professionals (CVcare), Institute for Health Services Research in Dermatology and Nursing (IVDP), University Medical Center Hamburg-Eppendorf (UKE), 20246 Hamburg, Germany; su.steinke@uke.de (S.S.); janna.lietz@gmx.de (J.L.); albert.nienhaus@bgw-online.de (A.N.); 2Faculty of Life Sciences, Hamburg University of Applied Sciences (HAW), 21033 Hamburg, Germany; z.schillmoeller@haw-hamburg.de; 3Department for Occupational Medicine, Hazardous Substances and Health Sciences (AGG), German Social Accident Insurance for the Health and Welfare Services (BGW), 22089 Hamburg, Germany

**Keywords:** health literacy, health behavior, state of health, vocational education, career-starters

## Abstract

(1) Background: The start of vocational education is a challenge for many people whose careers are just beginning. The working conditions exact new physical and mental tolls that can have an impact on their state of health and health behavior. Well-developed health literacy helps to encourage greater self-responsibility with respect to health and safety in the workplace. This study aimed to contribute to the evolution of health-related interventions in vocational training and instruction. (2) Methodology: This cross-sectional study examined health literacy, health behavior, and states of health among trainees engaged in work-and-study vocational training in 11 professions at the start of their education courses in northern Germany. The data were collected using a paper and pencil format. (3) Results: The survey was approved by 47 vocational schools (response rate 14%), with 1797 trainees returning their questionnaires (response rate 36%). The average age of the overall cohort was 21, and 70% of the trainees were female. A total of 47% of the participants began their careers with sufficient health literacy; health literacy was problematic in 40% of cases, and inadequate in 13% of cases. Around 50% of trainees exhibited poor dietary regime and risky alcohol intake, while 58% reported having a medical condition that had been previously formally diagnosed. (4) Conclusion: There is a need to provide support for developing a healthier approach to work at the start of vocational training.

## 1. Introduction

Health literacy (HL) provides resources and potential for enabling individuals to gain more control over their health and over factors that affect their health [1]. The importance of HL has resulted in the undertaking of many activities in the fields of research, government, and practical applications in recent years. The promotion of these skills has become part of political strategic documents such as the European Health 2020 conceptual framework [2] or the National Health Literacy Action Plan [3] for Germany. In a literature review, Sørensen et al. developed a definition of HL that was based on 12 conceptual models and 17 definitions [4]: “Health literacy is linked to literacy and entails people’s knowledge, motivation and competences to access, understand, appraise, and apply health information in order to make judgments and take decisions in everyday life concerning healthcare, disease prevention and health promotion to maintain or improve quality of life during the life course.”

HL in this context is understood to be an individual dynamic ability that can be learned. It also relates to the healthcare system and to the needs of that system. If external conditions or individual circumstances change, the acquired skills must be adapted and expanded accordingly [5].

HL influences well-being and health [6]. In the general population and also in different subgroups, associations of HL and indicators of health can be found in socioeconomically disadvantaged adults, patients, …, caregivers and their children and adolescents [7,8,9,10,11,12]. Since health behavior seems to be a mediator in the relationship between HL and health, there is also evidence in the literature on the relationship between HL and health behaviors [13,14,15,16]. International studies have demonstrated the HL deficiencies of the population in many countries [17]. Especially socioeconomic factors seem to determine limited HL. International studies show associations with low educational level, low social status, and migrant background or migrant experience [4,18,19,20]. Berens et al. found decreasing HL with increasing age in a German sample [21]. Limited HL is also associated with a poor state of health and poor health behavior, reduced participation in health activities, and greater usage of the healthcare systems [6]. In Germany, more than half of the population (54%) exhibit a limited HL [22]. In the recent HLS-GER 2 survey from 2020, this proportion increased up to 59% [20]. In this representative study, participants with limited HL described their state of health as being worse, consumed less fruit, and were less physically active than participants with sufficient HL.

Current concepts encouraging HL to promote a healthy working environment are designed to be employed in the workplace and in the education system [2,3]. There are around 1.3 million trainees in dual education in Germany in apprenticeships and vocational schools [23]. Around a quarter of vocational training agreements are terminated early each year (see reference above). The start of vocational education is a challenge for many people whose careers are just beginning. Work needs and work processes exact new physical and mental tolls and entail new social demands. These stresses are placed upon vulnerable young people who are still maturing into adults, who commonly do not believe that unhealthy behaviors will affect them adversely [24], and who are not yet broadly aware of occupational health and safety matters. These new demands can affect health behavior and their state of health. Studies have demonstrated the negative impact on dietary habits and the amount of physical exercise [25]. Likewise, health problems such as back pain or headache have been described in relation to work activities [26]. Good HL enables a person to be aware of and act in service of their health in relation to their occupation [3] and can help reduce work accidents and occupational diseases [6].

### Purpose and Problem

Measures for improving HL should be geared towards the life conditions and needs of the target group [1]. In the German vocational education system, HL has only been studied amongst university students to date [27]. To enable the development of measures in vocational education, it is first necessary to explain the occurrence of HL among trainees. To this end, 10 professions involving different physical stresses and with different gender distributions were selected, and popular professions in particular were also taken into consideration. The study population was expanded to include educators in vocational training as, much like employees in nursing, they are expected to teach health-related skills.

The vocational training courses come from five different industries: healthcare: geriatric nursing specialist, medical nursing specialist, medical assistant; cosmetic: hairdresser; education: educators; technical: plant mechanic for sanitary, heating and air conditioning equipment, electronics technician for operational and building systems; business and administration: office manager/management assistant, retail manager/assistant, wholesale and export manager/assistant, industrial clerk.

The aim of our study was to present (i) HL among trainees undertaking vocational training in a variety of professions, (ii) the distribution of HL relative to sociodemographic variables, and (iii) the distribution of HL relative to health behavior and the state of health at the start of vocational education.

The study presented here is part of a longitudinal study. Further surveys will examine the development of HL, health behavior, and states of health as well as the associations between HL and indicators of health. Additionally, satisfaction with the training experience over the course of vocational education and when trainees transition into actual employment will be examined in future surveys.

## 2. Materials and Methods

### 2.1. Study Design and Sample

The cross-sectional study for the start of vocational education was performed in Northern Germany. The performance of the study was approved by the education authorities to whom inquiries were submitted in the Federal States of Bremen, Mecklenburg-Western Pomerania, Lower Saxony, and Schleswig-Holstein; Hamburg was an exception in this regard (see Figure 1). An ethics endorsement was obtained from the Hamburg Medical Association (PV5670). The vocational schools were identified by searching the internet for the specified vocational training courses in the included federal states. All of the 321 identified vocational schools were contacted, with 47 agreeing to participate (response 14.6%). A total of 5052 trainees received the study documents from their teachers. The survey questionnaire and contact details for future surveys were returned by post to an independent anonymization center, which then forwarded the coded surveys to the office performing the analysis. A signed declaration of consent had to be provided for inclusion in the study cohort; minors were also required to present the consent of their parents or legal guardians. The data were collected between November 2017 and March 2018, with 1797 questionnaires eligible for inclusion in the analysis (response rate 35.5%).

### 2.2. Data Collection Instrument

The data were collected using a paper and pencil format in Spring 2018. Questions on sociodemographic data covered age, gender, country of birth, nationality, and highest level of educational attainment.

HL was surveyed using the short version of the validated German Health Literacy Survey Questionnaire (HLS-EU-Q16) [28]. This data collection instrument is based on the definition of HL quoted here. The 16 items in the fields of healthcare, disease prevention, and health promotion are used to collect information on the four skills relating to the handling of health-related information (access, understand, appraise, apply). The data were analyzed using the recommended procedures. Cases with values missed in more than two items were excluded beforehand. The four-stage response categories were binarized, and an aggregate score of 0 to 16 points (P) was calculated. This was used to create three HL level categories: sufficient (13–16 P), problematic (9–12 P), and inadequate (0–8 P).

Questions on state of health included a subjective assessment [25], medical conditions in the past 12 months [29], and mental well-being [30].

Regarding health behavior, data were collected on the frequency of physical exercise [31], smoking [32], alcohol consumption [33], and dietary habits [34].

On the basis of the state of health and health behavior, we categorized subjects into health lifestyles according to the work of Betz et al. [26]. The state of health is represented by 9 of 10 medical condition groups in the abridged version of the Work Ability Index. Accident-related injuries are excluded. Information was interpreted as a medical condition if “yes, diagnosed by a doctor” was marked. A dichotomous variable was used to differentiate between trainees without medical conditions and trainees with medical conditions. Health behavior was evaluated by dichotomizing the variables physical exercise, dietary habits, fast food consumption, smoking, and alcohol consumption (1 = healthy behavior, 2 = risky behavior) and aggregated into an index. Poor health behavior was deemed to be physical exercise of <2 h a week; poor dietary habits; the consumption of pizza, fries, and/or burgers daily or several times a week; smoking daily or occasionally; and risky alcohol consumption. The aggregate score of ≤7 resulted in allocation to “positive health behavior”, while an aggregate score of >7 resulted in allocation to “risky health behavior”. The cross-tabulation of state of health and health behavior enables differentiation between trainees who are healthy, have a positive health behavior, are potentially at risk, and who have risky health behavior.

A pretest was performed with trainees from four occupational groups.

### 2.3. Statistical Analysis

The sample was described in terms of absolute and relative frequencies (the latter specified as percentages), arithmetic mean values, and standard deviations. The individual items in HLS-EU-Q16 were presented as relative frequencies (percentages) and their 95% confidence intervals. To examine the relationships, we applied Spearman’s rank correlation coefficient and, for normal distributions, Pearson’s correlation coefficient, depending on the scale of measure. For unpaired group comparisons, the chi-squared test was applied for values with dichotomous outcomes. Data without normal distributions were analyzed using the Mann–Whitney *U* test and the Kruskal–Wallis test. The significance level was set at 5%. The software IBM SPSS Statistics for Windows, Version 23.0. (Armonk, NY, USA) was used for the statistical analyses.

## 3. Results

### 3.1. Sociodemographic Properties

Among the 1797 trainees, those striving to become medical assistants constituted the largest group at 17%. Those working towards becoming retail managers/assistants, educators, and office managers/management assistants each also represented more than 10% (see Table 1). The technical professions of plant mechanic and electronics technician had smaller shares of less than 5%. A total of 70% of the trainees were female. The average age was 21, with ages ranging from 14 to 53. Around 9% of participants did not have German citizenship.

Half of the trainees completed their school education with a “Realschule” qualification. In the German school system, all children attend elementary school (Grundschule) up to fourth grade. Afterwards, they are separated according to academic abilities and wishes of their families. They attend either Hauptschule, Realschule, or Gymnasium. The Hauptschule (grades 5–9) teaches the same subjects as the Realschule and Gymnasium, but at a slower pace and with some vocationally oriented courses. It leads to enrollment in a part-time vocational school. The Realschule (grades 5–10) leads to part-time vocational schools and higher vocational schools. The Gymnasium leads to a diploma called the Abitur and prepares students for university study or for a dual academic and vocational credential. Study participants who left school without any qualifications (1%) were mostly found in the hairdresser and electronics technician groups.

### 3.2. Health Literacy

Around 53% of the study participants had limited HL, with 40% possessing problematic HL and 13% inadequate HL (see Figure 2). A total of 47% of the trainees began their careers with sufficient HL, with the occupational groups hairdressers (57%), electronics technicians (55%), plant mechanics (51%), and geriatric nurses (51%) exhibiting shares of over 50%. Office managers/management assistants (34%) and industrial clerks (41%) exhibited the lowest shares of sufficient HL. These two groups also have the highest shares of participants with inadequate HL (office managers/management assistants at 21%, industrial clerks at 16%). There were significant differences between occupational groups (with weak effects), namely, in comparisons between office managers/management assistants and retail managers/assistants (*p* = 0.036), educators (*p* = 0.015), hairdressers (*p* = 0.005), and electronics technicians (*p* = 0.003). In the sample, there were no significant differences between the genders in terms of HL, and only very weak negative correlation with age (*r_s_* = −0.048, *p* = 0.043). Trainees who had left school without qualifications had significantly lower HL compared to trainees with Abitur qualifications (*p* = 0.020), Realschule qualifications (*p* = 0.009), Hauptschule qualifications (*p* = 0.001), and trainees with other school qualifications (*p* = 0.036). Trainees with Hauptschule qualifications have a much lower HL than those who possess qualifications from a vocational school (*p* = 0.024*)*. The effects in each case were weak. Trainees without German citizenship did not differ from other participants in terms of HL.

Responses to individual items regarding HL were highly diverse in terms of the proportions of persons responding to questions with “fairly difficult”/“very difficult” (see Table 2). While there were barely any difficulties in understanding the instructions of a doctor or pharmacist on how to take prescribed drugs (5%), one in two participants struggled to appraise whether information about health risks in the media was trustworthy (50%). The share of limited HL in terms of “appraising information” (57%) was higher than in terms of “accessing information” (48%), applying information (42%), and “understanding information” (21%). With a share of 48% in terms of limited HL, literacy relating to disease prevention was more poorly estimated than in relation to health promotion (43%) and healthcare (36%).

### 3.3. Health Behavior and Health Literacy

In all three HL groups, around one-third of the participants smoked daily (see Table 3). The proportion of smokers is significantly higher among men (*p* < 0.001) (see Figure 3). A total of 45% of female participants and 50% of male participants declared risky alcohol consumption—this proportion increased as HL decreased among women, while it decreased among men. Around one-third of the trainees engaged in two hours or more of physical exercise per week, men significantly more frequently than women (*p* < 0.001). As HL declined, so too did the proportion of trainees who engaged in physical exercise. A total of 52% of the trainees had an unhealthy diet, 29% had a normal diet, and 19% had a healthy diet, with the latter applying to women in particular. The proportion of participants with unhealthy diets was around 7% higher in the limited HL group than it was in the sufficient HL group.

### 3.4. States of Health and Health Literacy

A total of 37% of the trainees assessed their state of health as “very good” (see Table 3), with male trainees viewing their state of health significantly better than female trainees (*p* < 0.001) (see Figure 3). Over half of the participants (58%) began their training with at least one formally diagnosed medical condition. Female participants were significantly more frequently affected (*p* < 0.001). The most common conditions were disorders of the respiratory tract (18%), skin (16%), and musculoskeletal system (15%). Poor mental well-being was quoted by 44% of the respondents, with female study participants quoting this significantly more frequently than male respondents (*p* < 0.001). Trainees with sufficient HL had a more positive estimate of their state of health and reported a formally diagnosed medical condition or poor well-being less frequently than participants with limited HL (see Table 2).

### 3.5. Health-Related Lifestyles

One-quarter of the trainees had a conscious attitude towards their health and reported having no medical conditions (“healthy”) (see Table 3). Another 34% had a conscious attitude towards their health and were affected by at least one medical condition (“conscious”). A total of 17% of those with risky attitudes to health had not yet suffered a medical condition (“potentially at risk”), while 25% reported suffering from one or more medical conditions with a risky attitude to health (“at risk”). As HL diminishes, the proportion of trainees with a risky health-related lifestyle increases.

## 4. Discussion

We believe that the study presented here is the first to examine health literacy among trainees in a broad range of occupations. According to our data, 53% of respondents possess limited HL. Schaeffer et al. found in a recent study among the German population a proportion of 59% [20]. In a sample of youths aged 15 from Austria, 58% were found to have limited HL [28], while a study on HL among adults in Germany put this figure at 44% [35]. With a proportion of 46% with limited HL, a cohort from North Rhine-Westphalia, Germany, fell within the middle range of a ranking of the eight participating countries in a European comparative study, at a similar level to Greece and Poland. The level in the Netherlands was only 29% [17].

Another study examined HL among students at the University for Health Sciences in Bochum, Germany, wherein 69% of the students had limited HL [27]. Compared to the trainees in the healthcare professions, the proportion among the students was 15 to 20% higher. It is possible that the university students are driven to make a more critical assessment of their own health literacy due to their direct exposure to the science of health-related matters. On the other hand, it is also conceivable that participants in our study incorrectly overestimate their health literacy due to a lack of knowledge. Interestingly an investigation on adolescents aged 14–17 found out that the applicability of the long version of the HLS-EU-16 is limited [37]. Due to a high abstraction level, some participants found some questions to be too difficult. It would be conceivable that in our study these “challenges” biased the responses towards a sufficient HL.

There were no demonstrable gender-related differences in HL in our study or in the previously quoted studies [22,27,28,35]. While Schaeffer et al. demonstrated a high level of low HL among people aged 65 and over, the studies of Jordan/Hoebel and Reick/Hering found no significant age-related effects. Our study also only found a very weak effect of rising age on a decline in HL, which may have been related to the young age distribution of the sample. Among first- and later-generation immigrants, Schaeffer et al. found a high proportion of inadequate HL; there was no such correlation among the trainees in this regard. However, the state of being a first- or later-generation immigrant was only identifiable in our study using the nationality and country of birth variables, as it was not possible to collect information on parental origins due to data protection concerns. There was also a correlation demonstrated between limited HL and low education level among the trainees, affecting primarily participants who had left school without any qualifications—this was consistent with the findings of the studies of Schaeffer et al. and Jordan/Hoebel.

For the individual items of the HLS-EU-16, proportions for the responses “fairly difficult”/“very difficult” were calculated on the basis of the Jordan and Hoebel methodology. Compared to the results of a study on HL among adults [35], trainees mostly had much greater difficulty with finding and processing information. One reason for this may be that they have less experience with medical conditions; the fact that health-related networks are only just developing; and that when transitioning from healthcare by the family to a state of personal responsibility, there is a need to learn how to handle health-related matters. In comparison with the study among adults, there were considerable differences in items 13 and 16. Trainees found it more difficult to appraise which daily habits were related to their health (23% vs. 14%) and in accessing information about behaviors that are good for mental well-being (37% vs. 21%). Around 44% of the trainees reported a poor sense of well-being. The new demands coming along with vocational training may have a negative impact on mental health [38] and increase the need for information on this topic. If the deficiencies in well-being are related to the new demands, it can only be expected that this proportion will drop again as the vocational education progresses. The acquisition and processing of health-related information may be made more difficult in youth both by limited literacy and by inadequate access to appropriately presented information. Gray et al. illustrated that even experienced internet users only had limited benefits from online health information [39].

The higher prevalence of risky behaviors among youth [24] was confirmed in our study at all HL levels. Contrary to expectation, it was shown that there were no differences in smoking prevalence in the different health literacy groups. For men, there was even a positive correlation between health literacy and risky alcohol consumption. This result is inherently contradictory, as health literacy should have a direct impact on health behaviors. Zok/Böttger found that trainees with good health behavior were less prone to absenteeism than those who had a risky health behavior [38]. Despite a positive assessment of their own state of health in most cases, over half of the trainees at all HL levels began their vocational education with a medical condition that had been formally diagnosed.

To ensure that trainees who are just starting their vocational education can receive information on disease prevention and health promotion, it is first necessary to improve awareness of the relevance of health in the first place. It may be possible to encourage such awareness and make trainees more receptive with concepts related to well-being, enjoyment of life, and happiness [26,40]. When designing such services, the diverse nature of the group should also be taken into account, as illustrated by our results. Trainees with a positive attitude towards health can serve as role models—services should aim to motivate trainees to continue their positive approach to personal health [26]. For trainees at risk, on the other hand, prevention services are important to mitigate the problems of existing medical conditions and to encourage a more positive attitude towards health [25]. Effective networking between vocational schools and workplaces that provide vocational education placements may also enable long-term opportunities for the communication of health-related topics and help prevent sickness-related abandonments of vocational training.

Reviewing the literature, one does not find examples of interventions designed to increase HL of trainees in vocational schools. Peralta and Rowling reviewed studies that observed implementations of HL programs in Australian schools [41]. Only one of the three included studies was based on a theory-based health literacy framework, being the only successful study [42]. With respect to HL in the workplace, it also appears that theory-based interventions are promising. Larsen et al. show in their study on reduction of musculoskeletal complaints in the workplace that an intervention based on the framework of Jordan and Hoebel targets the relevant abilities (access, understand, appraise, and apply information) and improves HL significantly [43,44]. Thus, it makes sense to integrate a theory-based framework of HL into the development of an intervention improving HL of trainees, whether in the vocational school or in the workplace.

### Strengths and Weaknesses

To our knowledge, this is the first study in Germany reporting about HL in trainees. The cross-sectional design of the study means that it is not possible to draw conclusions on causal relationships. A longitudinal study design, however, would enable such conclusions for later data samples. According to our study protocol, two more follow-up assessments will be performed during training time and another two in subsequent working life. Although education authorities from four Federal States provided their support for the study, the response rate of the vocational schools was low, limiting the sample size. Therefore, the degree of generalization on the basis of the study results is limited. Despite the moderate response rate of the participants, it is not possible to eliminate the possibility of selection bias. The data protection requirements that education authorities are bound by prevented them from providing information on the trainees’ parents, which is why it was only possible to derive data on social status and on parental migration histories to a limited degree. The sample consisted of 70% women and 30% men and was therefore not balanced. Interviews with teenagers have suggested that there were problems with understanding the language and content of the long version of the HLS-EU-Q [37]. These also relate to items in the short edition, which may have affected our study results. Furthermore, the comparability of study results with studies in literature generated with the long version of the HLS-EU-Q might be affected.

## 5. Conclusions

Communicating and encouraging health literacy is important for the study group under review, as the majority of them showed limited HL. Raising awareness of the need to take a healthier approach to working life and provide trainees with the skills they need to be aware of their health in an occupational context may mitigate the development of existing medical conditions, reduce risky behaviors, and have a positive effect on well-being. Our findings show that there is no gender-specific difference in HL. In contrast, we observed gender-specific differences in health behavior and health outcomes. Thus, independent of HL measures for the total group “in order to make judgments and take decisions in everyday life concerning healthcare, disease prevention and health promotion…” [4], health promotion and prevention measures should account for gender differences in this group of trainees. Further qualitative research should be conducted to shed light on the sometimes-contradictory relationship between health behavior and health literacy among trainees. It is possible that the health literacy questionnaire is prone to misperceptions among adolescents, meaning that members of this group might not be able to recognize their lack of ability towards health literacy. Since the HLS-EU-Q does not collect data in an objective way like other HL instruments [45], but rather assesses subjectively rated HL, misclassification as a reason for contradictory results cannot be excluded. All in all, more research on the health literacy of trainees is needed to confirm the findings identified here.

## Figures and Tables

**Figure 1 healthcare-09-00757-f001:**
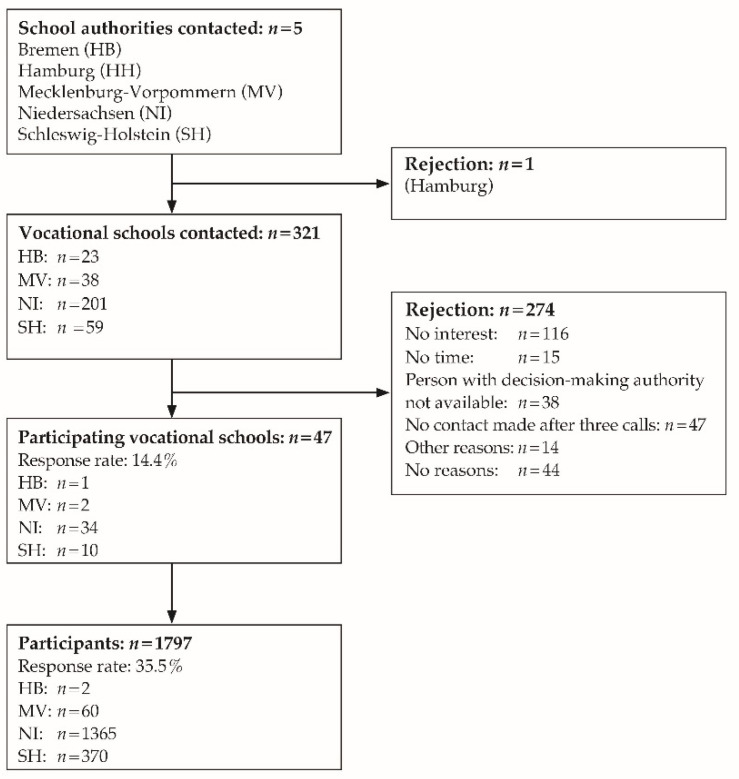
Recruitment process flow diagram.

**Figure 2 healthcare-09-00757-f002:**
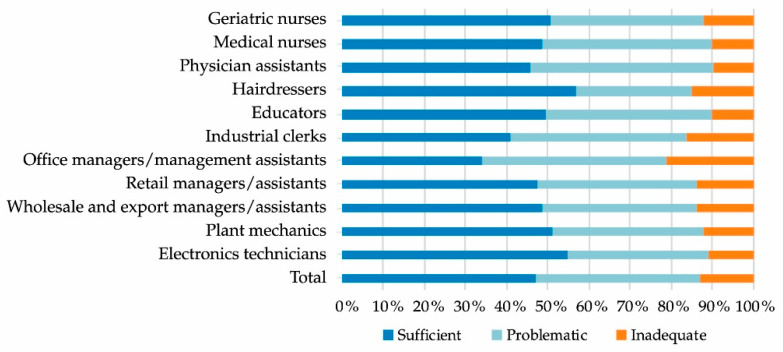
A sectoral comparison of health literacy (*n* = 1754).

**Figure 3 healthcare-09-00757-f003:**
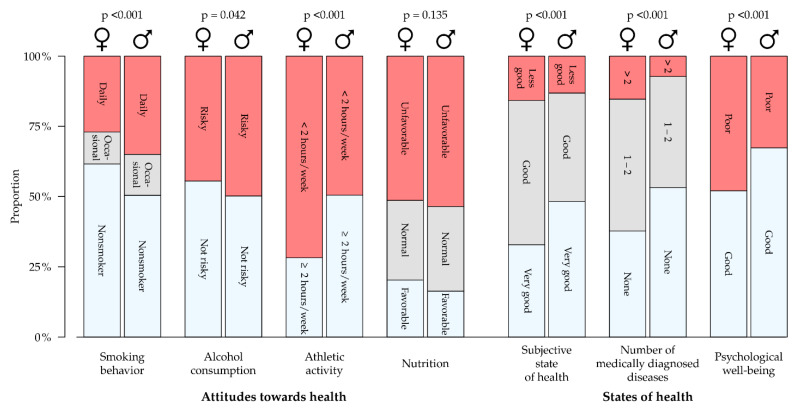
Health behavior and states of health according to gender.

**Table 1 healthcare-09-00757-t001:** Sociodemographic properties of study population.

Study Population *n* = 1797	Participants in Occupational Groups	Gender	Age	Other Nationality	School Qualifications			
					Hauptschule	Real-Schule	Fachhoch-Schule	Abitur
Missing value (*n*)		6	10	27	11			
	*n* (Valid %)	*n* Female (Valid %)	MW (SD)	*n* (Valid %)	Valid %			
Geriatric nursing	138 (7.7)	108 (78.8)	24.3 (7.6)	28 (20.7)	15.6	57.8	7.4	16.2
Medical nursing	139 (7.7)	124 (89.2)	22.5 (5.9)	13 (9.6)	0.0	42.5	12.9	44.6
Medical assistant	309 (17.2)	302 (98.1)	20.5 (4.4)	29 (9.5)	3.9	66.5	14.7	13.7
Hairdresser	117 (6.5)	96 (82.1)	20.8 (5.6)	28 (24.6)	48.7	36.5	1.7	3.5
Educator	246 (13.7)	206 (83.7)	22.6 (5.1)	3 (1.2)	0.0	54.6	28.2	16.3
Industrial clerk	116 (6.5)	52 (45.2)	20.0 (2.0)	4 (3.5)	0.0	13.8	21.6	63.8
Office manager/management assistant	182 (10.1)	138 (75.8)	21.1 (4.6)	8 (4.4)	1.7	49.7	24.9	23.2
Retail manager/assistant	260 (14.5)	161 (62.4)	20.6 (3.1)	23 (8.9)	33.2	54.4	6.6	4.3
Wholesale and export manager/assistant	163 (9.1)	67 (41.1)	20.5 (3.1)	4 (2.5)	0.6	36.2	22.7	36.8
Plant mechanic	49 (2.7)	4 (8.3)	20.8 (4.8)	5 (10.9)	52.1	27.1	6.3	12.5
Electronics technician	78 (4.3)	3 (3.8)	20.5 (4.5)	11 (14.5)	15.4	60.3	3.9	12.8
Total	1797 (100)	1261 (70.4)	21.3 (4.9)	156 (8.8)	12.1	49.4	15.3	20.9

**Table 2 healthcare-09-00757-t002:** Individual items in the HLS-EU-Q16 in relation to the relative frequencies (percentages) for the responses “fairly difficult”/“very difficult” based on Jordan and Hoebel [35]. English translation according to Sørensen et al. [36].

Item No.	Domain	Ability	On a Scale from Very Easy to Very Difficult, How Easy Would You Say It Is to…	% (95%-Confidence Interval)
**1**	**Healthcare**	Access information	…find information on treatments of illnesses that concern you?	20.3 (18.4–22.2)
**2**		Access information	…find out where to get professional help when you are ill?	16.3 (14.6–18.0)
**3**		Understand information	…understand what your doctor says to you?	15.5(13.8–17.2)
**4**		Understand information	…understand your doctor’s or pharmacist’s instruction on how to take a prescribed medicine?	5.0 (4.0–6.0)
**5**		Appraise information	…judge when you may need to get a second opinion from another doctor?	43.7 (41.4–46.0)
**6**		Apply information	…use information the doctor gives you to make decisions about your illness?	31.5 (29.3–33.7)
**7**		Apply information	…follow instructions from your doctor or pharmacist?	6.8 (5.6–8.0)
**8**	**Disease Prevention**	Access information	…find information on how to manage mental health problems such as stress or depression?	39.9 (37.6–42.2)
**9**		Understand information	…understand health warnings about behavior such as smoking, low physical activity, and drinking too much?	9.8 (8.4–11.2)
**10**		Understand information	…understand why you need health screenings?	11.5 (10.0–13.0)
**11**		Appraise information	…judge if the information on health risks in the media is reliable?	49.8 (47.5–52.1)
**12**		Apply information	…decide how you can protect yourself from illness based on information in the media?	44.2 (41.9–46.5)
**13**	**Health Promotion**	Access information	…find out about activities that are good for your mental well-being?	36.9 (34.6–39.2)
**14**		Understand information	…understand advice on health from family members or friends?	12.1 (10.6–13.6)
**15**		Understand information	…understand information in the media on how to get healthier?	29.1 (27.0–31.2)
**16**		Appraise information	…judge which everyday behavior is related to your health?	23.1 (21.1–25.1)

**Table 3 healthcare-09-00757-t003:** Attitudes to health, states of health, and health-related lifestyles grouped according to health literacy level.

Variable	Expressions	Health Literacy Sufficient	Health Literacy Problematic	Health Literacy Inadequate	Total
		Proportions as %			
Smoking habit	Non-smoker	58.0	59.7	56.8	58.3
	Occasional	11.6	12.3	13.7	12.4
	Daily	30.4	28.0	29.5	29.3
Risky alcohol consumption	Women	42.7	45.7	48.1	44.5
	Men	52.8	50.0	34.9	49.8
Physical exercise	<2 h/week	62.4	67.1	70.4	65.2
Diet	Healthy	21.7	17.0	16.2	19.2
	Normal	30.6	27.6	28.8	29.1
	Unhealthy	47.7	55.4	55.0	51.7
Subjective state of health	Very good	44.5	33.1	18.8	36.8
	Good	45.4	51.3	48.2	48.1
	Not very good	10.1	15.1	33.0	15.1
Number of medically diagnosed conditions	None	44.8	38.0	42.6	41.8
	1–2	45.6	46.2	39.9	45.1
	3–4	8.0	13.2	13.6	10.8
	≥5	1.6	2.6	3.9	2.3
Mental well-being	Poor	35.2	50.5	56.6	44.0
Health-related lifestyle ^a^	Healthy	26.4	19.9	28.1	24.0
	Conscious	33.5	37.3	27.2	34.3
	Potentially at risk	18.0	17.3	12.9	17.0
	At risk	22.1	25.5	31.8	24.7

^a^ Categorization according to Betz et al. [26].

## Data Availability

Data are available on request due to restrictions, e.g., privacy or ethical. The data presented in this study are available on request from the corresponding author. The data are not publicly available due to the fact that this was not subject to the informed consent.

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
