# Peer review of "Health Literacy, Health Behavior and States of Health among Trainee Personnel in Northern Germany"

_healthcare, 2021, doi:10.3390/healthcare9060757_

Round 1
Reviewer 1 Report
referring to :
line 261/262 On th other hand, it is also conceivable that participants in our study incorrectly overestimate their health literacy due to a lack of knowledge.
Thanks, it is good you name this here. The participants answer optimistic, especially, when it comes to judgement if Information on health risks given in the media is reliable (49,8).

Reviewer 2 Report
An excellent paper! Very well structured and significant contribution to health literacy knowledge as the implications for younger generations and the need for them to understand and manage their own Health Literacy is an important learning and development issue.
The research is supported by high participation numbers and the ethics processes and survey methodology have been well planned and presented and provides a strong outcome for the research team.
Referencing is contemporary, relevant and supports the research work.
Reviewer 3 Report
- Concerning the introduction, I would suggest revising it. In fact, at the beginning of this section, the focus is on the role of HL, its influence on health and well-being and the factors that determine a low level of HL. Only then a definition was provided. I would suggest giving the definition of the constructs at the beginning and only later its role. At the end of the introduction the focus is on the relationship between HL and health behavior and state of health. I would suggest exploring more in-depth health behavior and state of health and their relationship with HL, with more details and example, because importance was given to these constructs in the title and in the abstract. I would suggest expanding literature with more recent studies so that it can be created a dialogue between results and more recent literature.
- The authors investigate the occurrence HL among trainees because HL has been studied only among university students. In addition, the authors stated that, “measure for improving HL should be geared towards the life condition and the needs of the target group”. For this reason, I expected to find some ideas of interventions to be developed in line with the results of the analysis, but it’s not. The only idea present in the conclusion is “with regard to preventive measures improving health behavior and health status of trainees, gender-specific intervention should be implemented”, but in the results the authors stated, “there were no demonstrable gender-related differences in HL in our study or in the previously quoted studied”. I would suggest providing an explanation of this statement to understand if there are distinctions and specifics to do.
- Concerning the sample, the 70% of the sample are female and only 30% are male. For this reason, the sample seems to be unbalanced. I would suggest treating it in the limits.
- In the results, the authors used German qualifications, but they are not clear to a person from a different culture. I would suggest giving a definition and an explanation of this qualifications.
- In the conclusion, the authors stated, “It is possible that the health literacy questionnaire is prone to misperceptions among adolescents, that means that members of this group might not be able to recognize their lack of ability towards health literacy”, but in this study the average age was 21 and participants are no longer teenagers. I would suggest explaining this statement more in depth.
- From line 280 to line 295 I would suggest revising this statement because it is not immediately clear
- Concerning the conclusion, I would suggest revising it and giving more ideas of measures in line with the results and with the target of the study.
Round 2
Reviewer 3 Report
I consider your additions and explanation appropriate.
This manuscript is a resubmission of an earlier submission. The following is a list of the peer review reports and author responses from that submission.
Round 1
Reviewer 1 Report
It is considered a meaningful paper which it analyzes the relationship between Health Literacy, Health behavior and States of Health. In particular, the number of samples is large, and meaningful analysis results are expected to be drawn.
However, revisions are required for the following points.
First, the current analysis focuses on the relationship between the two variables, focusing on descriptive analysis. It is recommended to set independent and dependent variables and then do regression analysis. It is necessary to empirically verify the causal model by setting health literacy and other demographic variables as independent variables, and using health behavior or states health as dependent variables. Please refer to the Health Belief Model to review for the sake of adding new independent variables.
Second, to do causal analysis, it is necessary to derive a hypothesis. It is recommended to derive research hypotheses after reviewing the existing theoretical arguments.
Third, it is necessary to add the theoretical implications of this study. In addition, it is recommended to present topics that need additional research in future.
Reviewer 2 Report
Thank you for submitting your paper. The article provides an interesting analysis of the evolution of health-related interventions in vocational training and instruction in Germany.
Comments on individual parts:
- The abstract is precise enough.
- The keywords are correct.
- The Introduction should highlight not only the research problem and define the research gaps, but also should provide a brief description of the content of each section of the paper (in the last paragraph).
- The literature review presents important highlights on the state of the art. The major topics received enough attention and explanation.
- The use of research methods is adequate.
- The result section includes data analysis. The key findings can be considered significant and important. However, the critical assessment of the results will improve the quality of the analysis.
- The discussion of results is presented in a clear and logical way. However, the subsection “Strengths and Weaknesses” should be moved to Conclusions.
- The conclusions section should include a synthetic overview of the key research results as well as research limitations and potential directions for further research. This section is not well developed and does not include all the required information.
- Furthermore, it is necessary to refer to more recent literature. It would be valuable if you could use more references.
The academic language is correct, however, general proofreading would be advisable.
Reviewer 3 Report
I consider the issue of health literacy interesting and fairly original. I appreciate the survey and I consider the redemption rate good. Maybe it could be useful to strengthen the framework in the introduction and to have some comments in the conclusions. I mean to try to answer the question "so what?". Which are the policy and management implication of your analysis?
Round 2
Reviewer 1 Report
The core of the comment in the first review round is causal analysis. If there is not no revision about, the value of this paper is inevitably low.